# Effect of Climatic Condition, Type of Trough and Water Cleanliness on Drinking Behavior in Dairy Cows

**DOI:** 10.3390/ani14020257

**Published:** 2024-01-13

**Authors:** Franziska Katharina Burkhardt, Jason Jeremia Hayer, Céline Heinemann, Julia Steinhoff-Wagner

**Affiliations:** 1TUM School of Life Sciences, Technical University of Munich, 85354 Freising-Weihenstephan, Germany; f.burkhardt@tum.de; 2Institute of Animal Science, University of Bonn, 53115 Bonn, Germany; j.hayer@neumuehle.bv-pfalz.de (J.J.H.); c.heinemann@uni-bonn.de (C.H.); 3Educational and Research Centre for Animal Husbandry Hofgut Neumuehle, 67728 Münchweiler an der Alsenz, Germany; 4HEF World Agricultural Systems Center, Technical University of Munich, 85354 Freising-Weihenstephan, Germany

**Keywords:** livestock, water consumption, drinking water quality, trough cleaning, ambient temperature

## Abstract

**Simple Summary:**

Livestock water requirements are expected to triple. Simultaneously, freshwater availability declines while the risk of waterborne diseases rises with increasing temperatures. These challenges demand water management strategies on farms. Analyzing specific drinking behavior indicators can provide the basis for an improved water supply for dairy cows. This study aims to analyze drinking water quality and dairy cow drinking behavior in different climatic conditions, considering trough type and water cleanliness, and thereby optimizing water supply management for dairy cows. Dairy cows’ water supply is influenced by trough types, cleaning interval and cold and warm ambient temperatures, as shown by established drinking behavior indicators, particularly water intake and drinking breaks in number and duration, number of sips, as well as agonistic behavior. Considering this, animal welfare, especially freedom of thirst, can be improved.

**Abstract:**

Increasing ambient temperatures lead to higher water intake and higher risks of microbial growth in cattle troughs. This study aims to analyze drinking water quality and dairy cows’ drinking behavior (*n* = 8081 drinking episodes) on a commercial farm with 135 and 144 lactating cows in two climatic conditions, considering trough type and cleanliness, respectively. Daily video recording was conducted at two trough types (two open troughs, 70 L; two-valve troughs, variable volume of 5–15 L) in the first two hours after feeding (*n* = 60 days in total) under cold (December 2019–February 2020) and warm ambient temperatures (September 2021). The trough cleaning scheme allowed cows to access either cleaned or uncleaned troughs in each system. Water quality was tested daily and analyzed at the beginning and end of the trials. In warmer ambient temperatures, fewer and—at uncleaned troughs and open troughs—shorter drinking episodes were recorded, with longer but fewer water intake periods, longer drinking breaks, and fewer sips (*p* < 0.0001). Considering the drinking episodes, respectively, water intake and drinking breaks in number and duration, the number of sips and the number of agonistic behaviors might optimize dairy cow water supply and hygiene management.

## 1. Introduction

Climate change increasingly jeopardizes the availability of freshwater due to rising temperatures and extreme weather conditions; at the same time, scientists expect the water requirements of livestock to triple [1,2]. To address potential future challenges, it will be necessary to develop strategies focusing on livestock coping mechanisms and water management at the regional and farm levels [3,4].

Not only are dairy cows physically affected by high ambient temperatures and higher solar radiation, but they are also highly affected by the bacterial burden of water troughs and, thereby, the water quality [5,6]. Microbial growth in livestock water troughs, e.g., *Escherichia coli* or antibiotic-resistant bacteria, is known to increase with rising ambient temperatures [7]. Nevertheless, studies on water quality, considering different risk scenarios, are rare. Water intake is a potential indicator of a dairy cow’s health and temperature [8]. Characterizing and analyzing specific drinking behavior parameters can provide the basis for an improved water supply for dairy cows [9]. Several influences on drinking behavior have already been analyzed [10], such as levels of fecal contamination [11], the preference for standing or flowing water [12], treated or untreated water [13] or different trough volumes [14]. Most studies only include drinking behavior indicators that are automatically monitorable but potentially exclude more reliable behavioral indicators for assessing animal welfare. In this context, Rushen et al. [15] warned against relying exclusively on behaviors that can be evaluated automatically in the current state of technology. Water supply assessments demand a broad spectrum of behavioral patterns to interpret signs accurately. However, little information has been provided on how dairy cow drinking behavior changes depending on trough type, cleanliness and climatic conditions [10]. The evaluation of the effects of variables—such as climatic conditions—on dairy cow drinking behavior should be conducted along with other established animal- and management-related factors, as drinking behavior is complex [10,16]. In a previous study, which was conducted under a similar study design but in cold ambient temperatures (average water temperature: 10.8 ± 3.0 °C), we could demonstrate the effect of trough type, cleanliness and interaction on dairy cow drinking behavior and the biological drinking water quality. Rapid testing of the water ATP value is recommended as a useful farm hygiene indicator for soiling or bacterial growth and, therefore, was used to monitor the biological water quality [17]. Hence, this study aims (1) to identify potential key parameters of dairy cows’ drinking behavior for a potential automated evaluation of water supply management on dairy farms. This requires the manual generation of a sufficiently large learning data set (2), thereby evaluating the effect of different trough types, trough cleaning intervals and climatic conditions (cold ambient temperatures vs. moderate warm ambient temperatures), on drinking behavior and water quality in dairy cows [10]. Therefore, the data from the previous study were used for comparison to investigate the complementary effects of dairy cows’ drinking behavior at different trough designs and different cleaning intervals dependent on cold (previous study [10]) and moderately warm ambient temperatures (current study). We hypothesized that neglected trough cleaning in warmer conditions would increase the bacterial burden in uncleaned troughs and, thus, affect drinking behavior. We expected that more total drinking episodes would be recorded at warmer ambient temperatures and at cleaned troughs than at cold ambient temperatures and uncleaned troughs, and more agonistic interactions would occur. We further expected that cows would use open troughs more frequently than valve troughs.

## 2. Materials and Methods

The experimental facility and the methods used for drinking behavior analysis, sample collection and testing were previously published in detail by Burkhardt et al. [10].

### 2.1. Experimental Procedure

On a commercial farm, the lactating dairy cow herd’s drinking behavior was analyzed twice over 15 consecutive days dependent on (1) two different trough types (open troughs, length: 2.00 m; width, 0.43 m; depth, 0.15 m; volume, 70 L and double-valve troughs: length, 0.73 m; width, 0.32 m; depth, 0.10 m; variable volume, 5–15 L), (2) two different cleaning intervals (cleaned daily and uncleaned) and (3) two different climatic conditions: cold conditions with low ambient temperatures, which provides less risks for bacterial growth (December 2019 and February 2020) and a higher-risk at moderate warm ambient temperatures, from now on referred to as “warm ambient temperatures” (September 2021) (*n* = 60 trial days with four cameras resulting in total of 480 h video material) in the northern hemisphere. Due to technical limitations, individual animal identification was not possible. Since time-lapse cameras were used, the analyzed sequences are technically called photo sequences; for ease of understanding, they are referred to as video recordings below. Technical details of video recording, trough type, cleaning interval, water sampling and water quality analysis are provided by Burkhardt et al. [10].

### 2.2. Experimental Facility

The dairy cow herd (Holstein-Frisian cows) was kept in a symmetrical 45 m × 24.3 m free-range barn on a commercial farm in North Rhine Westphalia, Germany (195 m a.s.l., average annual temperature: 11.3 °C; average annual rainfall: 66 mm). Details of the housing conditions of the dairy cow herd were previously described by Burkhardt et al. [10]. The number of animals and dairy cow herd performances were altered to a small extent (Table 1). However, the number of animals in the herd remained constant during each evaluated trial. No animals were excluded during the trials. No intervention or treatment was administered to the animals during the study. The composition of feedstuffs as well as the supplied water were analyzed (Appendix A).

### 2.3. Analysis of Drinking Behavior, Water Quality and Climatic Condition

Drinking behavior was described using TLC 200 time-lapse cameras by Brinno (Taipei City, Taiwan). Image sequences were analyzed for 13 behavioral parameters visualized in Appendix A according to Burkhardt et al. [10] using the BORIS software (version 7.9) by Friard and Gamba [18].

Water was sampled for biological (*E. coli*, total viable count (TVC)) and physicochemical analysis analogous to Burkhardt et al. [10] to ensure comparability. 

The same trained researcher scored the visual soiling (no soiling (1) to heavy soiled (3)) of livestock drinking water at each trough before daily sampling (free adenosine triphosphate content using 3M™ Clean-Trace™, 3M, Neuss, Germany; water temperature and water pH measurements according to Burkhardt et al. [10].

Weather data loggers recorded every 10 min ambient temperature and humidity, which were summarized per day (Table 2) [10].

### 2.4. Statistical Analysis

Categorial variables were analyzed using R version 4.1.1, and continuous data (means ± standard error) were calculated using the MEANS procedure of SAS (version 9.4, SAS Institute Inc., Cary, NC, USA), according to Burkhardt et al. [10]. After testing the normality of variables, a statistical model comprising the independent variables trough type (open and valve trough), cleaning interval (cleaned or uncleaned) and climatic condition (cold and moderate warm), on dependent variables related to drinking behavior (i.e., duration of drinking and tasting, duration and number of water intake periods and drinking breaks, sips per drinking episode) of dairy cows was fitted. As an ad hoc decision, we used two models with two fixed factors each: climatic condition and trough type (Table 3) and climatic condition and cleaning status (Table 4) instead of a combined model with three possible fixed factors (climatic condition, trough type and cleaning interval) because we had access to climatic data long after the first statistical analysis for the first published work [10]. The effect of climatic conditions on categorical behavioral variables (only tasting, motions of tasting, swallowing difficulties, agonistic behaviors, interruptions due to agonistic behaviors) was analyzed by comparing the obtained odds ratios numerically, as interactions have previously been shown to be important to consider for dairy cow drinking behavior [10].

## 3. Results

### 3.1. Effects of Climatic Conditions on Dairy Cows’ Drinking Behavior

A total of 8081 drinking episodes were observed. Fewer drinking episodes were observed in warm ambient temperatures (*n* = 3978) than in cold ambient temperatures (*n* = 4103). In both conditions, most drinking episodes during the observation period occurred between 30 and 60 min after providing fresh TMR. An effect of climatic conditions on the total duration of drinking episodes was observed only through the interaction of factors. Shorter drinking episodes were recorded at warm compared to cold ambient temperatures at uncleaned troughs and open troughs (*p* < 0.01 and *p* = 0.08). Cows displayed longer but fewer water intake periods (*p* < 0.0001), longer drinking breaks (*p* < 0.0001) and fewer sips (*p* < 0.0001) in warm than cold ambient temperatures.

Dairy cow drinking behavior differed depending on the interaction of climatic conditions and trough type (Table 3, Figure 1) and the interaction of climatic conditions and cleaning interval (Table 4, Figure 1).

### 3.2. Effect of Climatic Conditions and Trough Type on Drinking Behavior

During both climatic conditions, cows visited more frequently open troughs than valve troughs (Table 3). The climatic condition and trough type affected the likelihood of “smelling”, “tasting by using the tongue”, “only tasting”, “agonistic behaviors” or “interruptions due to agonistic behaviors”, and “swallowing difficulties” (Figure 1).

### 3.3. Effect of Climatic Conditions and Cleaning Interval on Drinking Behavior

Dairy cows drinking behavior differed depending on climatic conditions and cleaning interval (Table 4, Figure 1). In warm ambient temperatures, cows drank longer at cleaned than uncleaned troughs and shorter at uncleaned troughs than those in cold ambient temperatures (*p* < 0.001). At cleaned and uncleaned troughs, fewer but longer drinking breaks and periods of water intake were observed, and fewer sips were recorded than in cold ambient temperatures (*p* < 0.001) (Table 4). The climatic condition and trough cleaning status affected the likelihood of “smelling”, “tasting by using the tongue”, “only tasting”, and “swallowing difficulties” (Figure 1).

### 3.4. Climatic Condition-Specific Effect on Water Quality

#### 3.4.1. Biological Water Quality

In total, the ATP content of the trough water fluctuated more and was significantly lower in daily cleaned troughs than in those uncleaned (Figure 2, Table 5).

Water ATP content ranged higher in warm ambient temperatures than in cold ambient temperatures (Δ 2.08 log_10_ RLU/mL vs. Δ 2.86 log_10_ RLU/mL, respectively).

In warm ambient temperatures, the water was more frequently “soiled” (29%) and “heavily soiled” (25%), whereas in cold ambient temperatures, these ratings were recorded less frequently (16% and 9% of the daily measurements, respectively). The rating “clean” was recorded in 45% of the measurements at warm ambient temperatures and 75% of the measurements at cold ambient temperatures.

The microbiological analysis of livestock drinking water showed, on average, numerically higher CC and TVC counts in warm ambient temperatures than in cold ambient temperatures. In warm ambient temperatures, the average CC and TVC at 20 °C were slightly above the recommended reference values in uncleaned troughs. The average TVC incubated at 36 °C in all water samples from the troughs under both climatic conditions was higher than the reference values (Appendix A).

#### 3.4.2. Physicochemical Water Quality and Temperature

Physicochemical trough water quality was according to recommendations for livestock drinking water quality (Appendix A) [19]. The water temperatures were significantly higher in warm than in cold ambient temperatures in all troughs (Table 2).

### 3.5. Influence of Climatic Conditions and Water Quality Parameters on Drinking Behavior

Climatic measurements were significantly higher in warm than in cold ambient temperatures (Table 2).

Environmental conditions and water quality affected five drinking behavior parameters (Table 6).

## 4. Discussion

Cows increase their daily water intake with increasing ambient temperatures to maintain homeostasis [20]. In contrast to this finding and the corresponding initial hypothesis of the current study, fewer drinking episodes were recorded during warm than during cold ambient temperatures. The number of animals was slightly higher in warm than in cold conditions. This difference in the number of drinking episodes between climatic conditions might be attributable to the design of our study. Drinking behavior in both climatic conditions was recorded for the first two hours after feeding (i.e., 09:00 h to 11:00 h). However, several studies have investigated dairy cow drinking frequencies according to climatic conditions to describe the diurnal rhythms of cows. Ray et al. [21] fed feedlot beef cattle at 07:00–08:00 h h and 15:00–16:00 h in July and August (17.8–45.0 °C) and 14:00–15:00 h in February and March (0.6–33.9 °C), corresponding to winter in the northern hemisphere; cattle consumed most of their daily water intake at approximately 23:00 h and 08:00 h in warm ambient temperatures and 10:00–13:00 h in cold ambient temperatures [21]. Cardot et al. [22] observed water consumption peaks of Holstein Friesian cows in three consecutive trials (November–April, 2.4–5.8 °C) from 09:00–11:00 h, 17:00 h or 19:00 h. Dado and Allen [23] found that the number of drinking bouts and their size were negatively correlated (r = −0.77). According to Laínez and Hsia [24], at warm ambient temperatures (25–36 °C), cows drank most between 10:00 h and 19:00 h, whereas in cold ambient temperatures (13–25.2 °C), cows drank most in the morning (fed at 05:30 h and 14:00 h). In this context, season and time of day interacted significantly. Furthermore, Holstein Frisian cows exhibited higher water intake at warm ambient temperatures than at cold ambient temperatures. Cows tendentially drank more when visiting the trough [24]. The time spent drinking in both climatic conditions was not different. Nevertheless, cows drank significantly more at warm (61.9 L per day) than at cold ambient temperatures (38.6 L per day) [24]. These findings suggest that the lower drinking episodes in warm ambient temperatures than in cold ambient temperatures might result from a shift in water consumption periods to cooler times of day in warm ambient temperatures and thereby out of the recording period in the current study.

Measuring water consumed per cow was not feasible in our study due to technical limitations. The number of sips taken per drinking episode could likely be used to estimate water intake since water expenditure also includes spilled water and might overestimate water intake. However, fewer sips were counted at warm ambient temperatures than at cold ambient temperatures. This contrasts with the regression analysis by Meyer et al. [25], indicating that, at a mean water consumption of 81.5 kg/cow/day, for each additional degree Celsius at ambient temperature, water consumption rose by 1.52 kg per day. Multiple studies demonstrate a positive correlation between ambient temperature and water intake [26,27]. Cardot et al. [22] observed cows during cold ambient temperatures; all visited the trough without actually drinking at least once, for a mean of 0.3 ± 1.1 times/d per cow. McDonald et al. [28] hypothesized that cows are “attracted to the cooling effect of water on their skin” with increasing temperatures. This would explain the observed climatic condition-specific differences in drinking episodes, from short, alternating periods of water intake and drinking breaks in cold ambient temperatures to more extended periods of water contact and longer periods of drinking breaks in warm ambient temperatures at both tank and valve troughs.

In our experiment, cows could choose between two identical large-volume open troughs (70 L) and two identical small-volume valve troughs (variable volume between 5 and 15 L). Previous studies have reported that the trough material [29,30], trough volume, and surface area affect dairy cow water intake [30]. Cows visited open troughs (*n* = 5365) more frequently than valve troughs (*n* = 2716). In warm ambient temperatures, cows drank more frequently (but shorter) at open troughs than at valve troughs, with higher likelihoods of agonistic behaviors and resulting interruptions. McDonald et al. [28] reported that increasing temperatures lead to competition over water resources. Open troughs allow up to four animals to drink simultaneously and provide a wider water surface, thus possibly provoking higher competition at the troughs. A few dominant cows could block the water source and cool their skin without the actual need for water intake; this would explain the lower number of sips and more extended periods of water intake, along with a higher likelihood of agonistic behaviors and interruptions at open troughs in warm ambient temperatures. The climatic condition and trough type did not significantly affects the tasting duration. However, in warm ambient temperatures, the likelihoods of “smelling” and “tasting with tongue” were lower for open troughs than valve troughs, possibly due to the greater soiling of open troughs caused by different cleaning intervals.

The current study also evaluated the potential effects of the trough cleaning interval and the associated biological water quality on dairy cow drinking [10]. Increasing temperatures lead to microbial growth in cattle water troughs [6,7]. In the present study, the numerically higher CC and TVC counts in warm ambient temperatures than at cold ambient temperatures, with CC and TVC values exceeding the reference values at 20 °C in warm ambient temperatures in the uncleaned troughs, support those findings. The current study observed a higher trough water ATP content at warm ambient temperatures in uncleaned troughs than in those cleaned daily. Trough water was also more frequently soiled at warm ambient temperatures than at cold ambient temperatures. In both climatic conditions, cows drank more frequently at uncleaned than cleaned troughs. This contrasts with previous studies reporting a preference for clean water over manure-contaminated water based on the water consumed by cattle [11,12]. However, cows spent more time in cleaned troughs at warm ambient temperatures. They took more sips during longer water intake periods, with a lower likelihood of smelling while tasting.

In contrast, cows took fewer sips at cleaned troughs in cold ambient temperatures, and the likelihood of “only tasting” was lower. These findings indicate that dairy cows’ preferences and sense of taste are reflected in different drinking behaviors, not only in the amounts of water consumed and drinking frequencies. Further results support this assumption: the higher the ATP measurements were, the lower the number and duration of drinking episodes, drinking breaks, water intake and the number of sips taken. These results indicate decreased acceptance of the drinking water provided as its ATP levels increase. The extent to which diseases are associated with dairy cow drinking water bacterial contamination remains unclear. Therefore, in the global problem of growing freshwater depletion and water deficits [31], the authors consider daily drinking water cleaning and daily water replacement sufficient but recommend systematically analyzing the biological water quality. Furthermore, the effects of possible taste-reducing factors such as odor [32], organic fractions in feces [33], water temperature [34] and chemical components [35,36] on the water intake of dairy cows are assumed to be amplified at warm ambient temperatures [10]. Regular assessments of water troughs, especially in warm ambient temperatures, are needed to account for these effects [7].

The current study indicates that open troughs are predominantly used for water consumption but also for cooling during warm ambient temperatures, characterized by longer water intake periods. The dual usage of and higher preference for open troughs increases agonistic behavior around open troughs, most likely resulting in decreased availability for low-ranking cows. The reduced availability is especially concerning given that more cows are classified as subordinate (64%) than dominant (27%) [33]. Most recommendations for water availability [11,19,37,38] are based on daily water demand and do not incorporate the additional usage of troughs for cooling; thus, these recommendations should be critically reviewed.

Recent studies comparing dairy cows’ water intake and drinking behavior in warm and cold climatic conditions are rare, and cross-study comparisons are limited. More studies on dairy cows’ water consumption have been conducted in cold climates (*n* = 7) than in warm climates (*n* = 2), and fewer studies have been conducted in cool climatic conditions (*n* = 2) than in hot climatic conditions (*n* = 7) [7]. Discussing the coping strategies of dairy cows under different climatic conditions requires a broader perspective, including other animal, management or resource-related stressors, as adaptation is determined by several factors [3].

## 5. Conclusions

Our study revealed substantial changes in dairy cows’ water quality and drinking behavior depending on climatic conditions, trough types and cleaning intervals. Warm temperatures amplified the effect of trough type and cleanliness on an impairment of water quality and different drinking behaviors. Useful behavioral indicators for further studies might be the frequency and duration of drinking episodes, water intake, drinking breaks, the number of sips and the frequency of agonistic behaviors. A better understanding of the palatability of water and tasting behavior could allow this behavior to serve as an indicator of sufficient water quality and optimize the management of limited drinking water resources. Future studies are needed to address how cows’ individual drinking behavior and water consumption are affected by the water quality of livestock drinking troughs.

## Figures and Tables

**Figure 1 animals-14-00257-f001:**
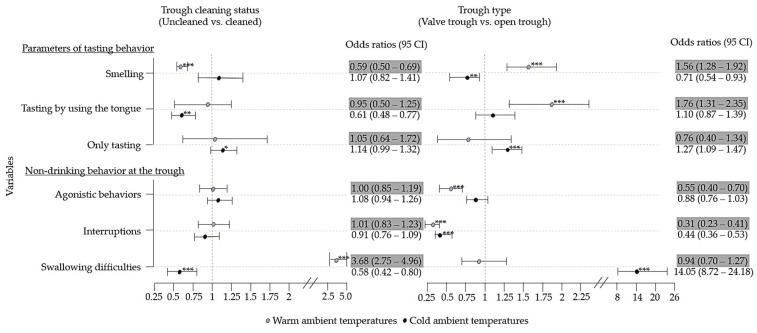
Effects of trough type and trough cleaning interval on dairy cows’ drinking behavior in two climatic conditions (warm ambient temperatures and cold ambient temperatures), visualized by odds ratios (dots and whiskers showing the 95% confidence intervals (CI)). * 0.05 < *p* < 0.1; ** *p* < 0.05; *** *p* < 0.01.

**Figure 2 animals-14-00257-f002:**
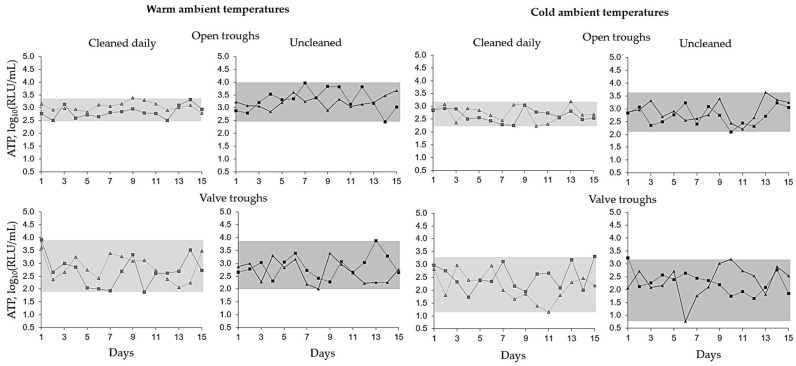
Adenosine triphosphate (ATP) content in four troughs (open trough 1 ▲, open trough 2 ∎, valve trough 1 ▲ and valve trough 2 ∎) in cold ambient temperatures and warm ambient temperatures. Troughs were cleaned daily or not cleaned for 15 days. The ATP range is shown in grey.

**Table 1 animals-14-00257-t001:** Performance traits of the experimental herd under cold ambient temperatures (December–February 2019/2020) and warm ambient temperatures (September 2020).

	Warm Ambient Temperatures (*n* = 135)			Cold Ambient Temperatures (*n* = 144)		
Variable	Min	Mean	Max	Min	Mean	Max
Days in milk (d)	14	198	609	10	170	470
Lactation number	1	2.3	9	1	2.2	8
Milk production (kg/cow and year)	11.1	32.2	57.9	13.7	32.8	57.0
Milk fat (%)	2.4	4.3	6.5	2.1	4.3	6.3
Milk protein (%)	2.6	3.65	5.0	2.8	3.6	4.4

**Table 2 animals-14-00257-t002:** Mean with standard error (SE) of the climatic conditions measured at all four of the experimental barns under two climatic conditions (warm and cold ambient temperatures) over two 15-day study periods for each condition.

Variable	Warm Ambient Temperatures		Cold Ambient Temperatures		
	Mean	SE	Mean	SE	*p-*Value
Minimum water temperature (°C)	12.1	1.3	3.1	0.9	<0.01
Maximum water temperature (°C)	20.8	0.5	16.2	0.5	<0.001
Mean water temperature (°C)	16.6	0.2	10.8	0.3	<0.001
Open troughs	17.3	0.2	11.6	0.3	<0.001
Valve troughs	16.0	0.3	10.1	0.4	<0.001
Mean ambient temperature (°C)	16.6	0.3	6.7	0.3	<0.001
Min	11.3	-	−0.3	-	-
Max	25.8	-	11.3	-	-
Relative humidity (RH) (%)	73.9	0.9	83.4	1.4	<0.001

**Table 3 animals-14-00257-t003:** Dairy cows’ drinking behavior was observed at two trough types for 15 d in two climatic conditions. Drinking behavior variables are shown as means with a standard error (SE). Significant differences (*p* < 0.05) between climatic conditions and trough types are highlighted by different letters.

	Warm Ambient Temperatures		Cold Ambient Temperatures		*p-*Value		
Variable	Open Troughs	Valve Troughs	Open Troughs	Valve Troughs	Climatic Condition	Trough Type	Climatic Condition
							×
							Trough Type
Drinking episodes (number)	*n* = 2912	*n* = 1066	*n* = 2453	*n* = 1650			
Drinking episodes (duration, s)	112.1 ± 1.7 ^B,b^	132.5 ± 3.3 ^a^	114.6 ± 1.6 ^A^	134.6 ± 2.5	0.08	<0.001	0.7
Tasting period (duration, s)	31.1 ± 0.8	35.4 ± 1.9	31.7 ± 0.8	33.7 ± 1.0	0.1	0.3	0.6
Drinking breaks (number)	2.2 ± 0.03	2.3 ± 0.1	2.7 ± 0.1	3.1 ± 0.1	n.a.	n.a.	n.a.
Drinking breaks (duration, s)	44.9 ± 1.1 ^A,b^	52.8 ± 0.1 ^A,a^	13.6 ± 0.3 ^B,b^	15.0 ± 0.5 ^B,a^	<0.001	<0.001	0.01
Water intake (duration, s)	57.2 ± 0.9 ^A,b^	71.5 ± 2.0 ^A,a^	26.5 ± 0.5 ^B,b^	27.2 ± 0.7 ^B,a^	<0.001	0.01	<0.001
Water intake periods (number)	2.4 ± 0.1 ^B^	2.4 ± 0.1 ^B^	3.1 ± 0.1 ^A^	3.1 ± 0.1 ^A^	<0.001	0.3	0.7
Sips (number)	12.2 ± 0.2 ^B^	12.3 ± 0.3 ^B^	20.2 ± 0.3 ^A,a^	19.7 ± 0.5 ^A,b^	<0.001	0.02	0.06

^A,B^ Differences in uppercase superscript letters indicate significant differences (*p* < 0.05) between ambient temperatures (warm and cold). ^a,b^ Differences in lowercase letters indicate significant differences (*p* < 0.05) between trough types (open trough and valve trough) within the same season; n.a. not available.

**Table 4 animals-14-00257-t004:** Dairy cows’ drinking behavior in warm ambient temperatures (*n* = 135 cows) and in cold ambient temperatures (*n* = 144 cows) over a period of 15 d were either daily cleaned or not. Drinking behavior variables are shown as means with a standard error (SE). Significant differences (*p* < 0.05) between climatic conditions and cleaning status are highlighted by different letters.

	Warm Ambient Temperatures		Cold Ambient Temperatures		*p-*Value		
Variable	Daily Cleaned	Not Cleaned	Daily Cleaned	Not Cleaned	Climatic Condition	Cleaning Interval	Climatic Condition
							×
							Cleaning Interval
Drinking episodes (number)	*n* = 1960	*n* = 2018	*n* = 1948	*n* = 2155			
Drinking episodes (duration, s)	123.1 ± 2.3 ^a^	112.2 ± 2.0 ^B,b^	119.3 ± 2.0	125.6 ± 2.0 ^A^	<0.01	0.2	<0.001
Tasting period (duration, s)	33.3 ± 1.2	31.5 ± 1.0	31.3 ± 0.9	33.6 ± 1.0	0.1	0.6	0.3
Drinking breaks (number)	2.3 ± 0.4 ^B,a^	2.1 ± 0.1 ^B,b^	2.8 ± 0.1 ^A^	3.0 ± 0.1 ^A^	<0.001	0.3	<0.001
Drinking breaks (duration, s)	49.7 ± 1.5 ^A,a^	44.4 ± 1.4 ^A,b^	13.7 ± 0.4 ^B^	14.6 ± 0.4 ^B^	<0.001	0.08	<0.001
Water intake (duration, s)	62.5 ± 1.2 ^A,a^	59.7 ± 1.3 ^A,b^	27.2 ± 0.7 ^B^	26.3 ± 0.5 ^B^	<0.001	0.02	<0.01
Water intake periods (number)	2.4 ± 0.1 ^B,a^	2.3 ± 0.1 ^B,b^	3.0 ± 0.7 ^A^	3.1 ± 0.1 ^A^	<0.001	0.08	<0.01
Sips (number)	12.6 ± 0.2 ^B,a^	11.9 ± 0.3 ^A,b^	18.6 ± 0.3 ^A,b^	21.3 ± 0.4 ^B,a^	<0.001	0.7	<0.001

^A, B^ Differences in uppercase superscript letters indicate significant differences (*p* < 0.05) between ambient temperatures (warm and cold). ^a, b^ Differences in lowercase letters indicate significant differences (*p* < 0.05) between cleaning intervals (daily cleaned and uncleaned) within the same climatic condition.

**Table 5 animals-14-00257-t005:** Adenosine triphosphate (ATP) content (log_10_ RLU/mL) of dairy cows’ drinking water in two climatic conditions under two different cleaning intervals, either daily cleaned or not.

Trough Cleaning Interval	Warm Ambient Temperatures	Cold Ambient Temperatures	Total	* p*-Value
Daily cleaned (log_10_ RLU/mL)	2.8 ± 0.05	2.5 ± 0.06	2.5 ± 0.03	*p* < 0.001
Not cleaned (log_10_ RLU/mL)	3.0 ± 0.05	2.6 ± 0.05	2.9 ± 0.03	*p* < 0.001

**Table 6 animals-14-00257-t006:** Spearman rank correlations between environmental conditions (ambient temperature and relative humidity), water quality parameters (water temperature and adenosine triphosphate) and dairy cow drinking behavior.

Variable	Correlated Parameter	r	*p-*Value
Ambient temperature	Drinking breaks (duration, s)	r = 0.3	<0.001
	Water intake (duration, s)	r = 0.3	<0.001
	Sips (number)	r = −0.2	<0.001
Relative humidity	Drinking breaks (duration, s)	r = 0.3	<0.001
	Water intake (duration, s)	r = 0.3	<0.001
Water temperature	Drinking breaks (duration, s)	r = 0.7	<0.001
	Water intake (duration, s)	r = 0.7	<0.001
	Sips (number)	r = −0.5	<0.001
Water ATP content	Drinking episodes (number)	r = −0.3	<0.001
	Drinking episodes (duration, s)	r = −0.2	<0.001
	Drinking breaks (duration, s)	r = −0.3	<0.001
	Water intake (duration, s)	r = −0.2	<0.05
	Sips (number)	r = −0.4	<0.001
Soiling of the trough water	Drinking episodes (duration, s)	r = −0.2	<0.001
	Drinking breaks (duration, s)	r = −0.2	<0.01
	Sips (number)	r = −0.3	<0.001
Trough water pH content	Drinking episodes (duration, s)	r = −0.3	<0.001
	Drinking breaks (duration, s)	r = 0.3	<0.001
	Water intake (duration, s)	r = 0.3	<0.001
	Sips (number)	r = −0.5	<0.001

## Data Availability

The data presented in this study are available on request from the corresponding author. The data are not publicly available due to further scientific use to establish AI.

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
