# Peer review of "Effect of Climatic Condition, Type of Trough and Water Cleanliness on Drinking Behavior in Dairy Cows"

_animals, 2024, doi:10.3390/ani14020257_

Round 1

Reviewer 1 Report

Comments and Suggestions for Authors

The topic can be of interest, and the authors are clearly expert on the topic. But I do not like a couple of things that I am not sure the authors are able to address:

1. I really struggle to see the value of two temperature conditions. I feel that these two categories does not reflect the full spectrum of temperatures (you have variations between -0.3 and 25.8°C) and also does not provide an exhaustive range of conditions to test potential climate change effects. I am wondering if it would make more sense to use temperature as continuous variable. I understand this split into two categories is done to simplify the outputs, but I am wondering if this way you limit your analysis. My understanding is that you also collected data over a limited time frame, so, as you discuss, your findings might be biased. Furthermore, I believe that the temperature variations between 9-11 are not so extreme as the rest of the day (it is not clear if the weather conditions are for the whole day or just 9-11, I guess the former, so data are not really associated to the right temperatures). It is also weird that you start your discussion with this big bias (good that you are not hiding it, but usually limitations are presented a bit later). There are some interesting data collected, but I think the narrative on climate change is really speculative. 

2. I do not understand how a single model can be too small putting despite more than 8000 observations. It does not make sense to have two separate models. Also it is not clear which type of model was used (it is not specified at all). Also you can add interaction effects in the model if you are interested in climatic*trough or climatic*cleaning status.

Considering these two limitations, I think the paper is not suitable for a Q1 journal such as Animals.  

Comments on the Quality of English Language

none

Reviewer 2 Report

Comments and Suggestions for Authors

Overall the writting of the introduction and abstract needs to be improved. The flow of the writing is hard to follow. I think the authors just need to change some sentences to make it clearer. Fro example, in 3.5 the authors show results for water ATP content and other biological water quality factors that are barely mentioned in the introduction. Although their evaluation is then explained in the materials and methods, the reader would benefit from a bit more information in the introduction. The biggest handicapp of the manuscript is the fact that the cows were not followed individually. This can deffinetely affect the results and should be explored in the discussion, as it was done for the times the authors chose to do the measurements. During discussion most results are discussed without using p-values. The authors should add them throughout the discussion section.

L48-52: This paragraph is a bit confusing since it starts with the relationship between milk and water intake by dairy cows and ends with radiation and coat color. The authors should rewrite this paragraph in order to address the hyphotesis of the manuscript.

L254: In this sentence, the last part does not connect with the first. 

L262: This information should be in the materials and methods.

L293: This is the type of sentence that would be helpful in the introduction and then could be dissected in the discussion.

L297: Could this have something to do with the troughs placement in the building? Or were they side by side with the clean ones?

Reviewer 3 Report

Comments and Suggestions for Authors

GENERAL APPRAISAL

The topic and information are interesting and novel to some extent. However, there are some issues to be considered before taking a decision.

Firstly, it seems that a previous paper published by these researchers contain a substantial amount of similar information with the only difference of not including environmental effects into the statistical model. I would suggest to the editorial office double-checking an overlapping of findings in the following published paper: Burkhardt, F.K.; Hayer, J.J.; Heinemann, C.; Steinhoff-Wagner, J. Drinking behavior of dairy cows under commercial farm conditions differs depending on water trough type and cleanliness. Appl. Anim. Behav. Sci. 2022, 256, 105752, doi:10.1016/j.applanim.2022.105752. If the overlapping is substantial, the manuscript should be rejected. If not, the manuscript under review should acknowledge that the findings were partially published.

On the other hand, the Editor In-Chief should weight the value of obtaining different findings by leaving out a variable in a previous article. As it is expectable, when including a new independent variable into the model, the results will change. In such a case, in order to save part of the information, a Solomonic decision could be asking to the authors to prepare a short communication solely containing results not colliding in any way with those previously reported.

TITLE, OBJECTIVES, RESULTS AND CONCLUSIONS

It is suggested to take care of having consistency among title, objectives, results and conclusions because it looks loose. The lack of consistency can be quickly evidenced when comparing what it is said across each segment:

Title: Optimizing dairy cows water supply by drinking behavior indicators.

Simple summary: This study aimed to analyze dairy cows drinking behavior in different climatic conditions, considering trough type and water cleanliness, and thereby optimizing water supply management in dairy cows.

Abstract: This study aimed to analyze dairy cows drinking behavior.

Introduction: This study aimed to identify potential key parameters of dairy cows drinking behavior for a potential automated evaluation of water supply management on dairy farms.

We hypothesized that warmer conditions would increase the impact of infrequent trough cleaning on drinking water quality (and thus drinking behavior) and that more total drinking episodes would be recorded in warmer ambient temperatures than at cold ambient temperatures. We expected that cows would use open troughs more frequently than valve troughs.

Results: Across stated results, subtitles read as follows:

3.1. Effects of climatic condition on dairy cows drinking behavior.

3.2. Effect of climatic condition and trough type on drinking behavior.

3.3. Effect of climatic conditions and cleaning interval on drinking behavior.

3.4. THERE IS NO 3.4. It seems that it was missed.

3.5. Climatic condition specific effect on water quality.

            3.5.1. Biological water quality.

            3.5.2. Physicochemical water quality and temperature.

3.6. THERE IS NO 3.6. It seems that it was missed.

3.7. Influence of climatic conditions and water quality parameters on drinking behavior. Moreover, alignment of 3.7 seems incorrect.

Conclusions: We found that dairy cow drinking behavior changed significantly according to climatic condition, trough type, and trough cleaning scheme. In particular, warm temperatures amplified the factors leading to different drinking behaviors observed in the evaluated trough types and trough cleaning schemes.

Thus, it is suggested to match title, objectives, results and conclusions. It seems to me that the easiest way to do it is using a new title such as: “Effect of climate, type of trough and water quality on drinking behavior in dairy cows”. 

ABSTRACT

OK

INTRODUCTION

OK

MATERIALS & METHODS

General impression: In general, this section looks limited and unclear. It is mainly because a substantial amount of information is referred to a previously published paper. From my stand point, referring a particular procedure or detail to a previous paper is feasible but not at so extensive level that makes impossible to repeat the experiment following the described M&M.  

P3/L82-83: Eventually, a researcher somewhere in the planet may be unable to find the referenced paper (Burkhardt et al., 2022. Appl. Anim. Behav. Sci. 256, 105752). Thus, at least briefly, I would suggest to describe location, type and number of animals, selection of animals (inclusion and exclusion criteria) and their housing, feeding and handling, etc.

P4/L125-136: Perhaps my ignorance, but I think that the experiment could not be repeated with the provided information regarding statistical procedures. Please explain which is the reason behind… “due to the study design” forcing to you to use 2 statistical models ?. Why not a Three way ANOVA ?. Was normality of variables tested ?.

It is suggested to clearly state statistical models as follows: “In order to test the effect of factors affecting water drinking behavior, a statistical model comprising the independent variables environmental temperature, type of trough and water cleanliness on dependent variables related to drinking behavior (i.e., number of drinking’s, duration of drinking episodes, smelling, tasting, number of sips) of dairy cows was fitted.

Description of response variables (e.g., drinking episodes, smelling, tasting, number of sips, etc.) should be provided.

Variable’s units are not mentioned.

P4/L133-134: Compared or similar ?. In such a case, similar results ?.

RESULTS

What is the difference between a sip and a drinking episode ?. What is the difference between drinking episodes and water intake periods ?. Not defined in M&M.

P5/L157: in terms of drinking episodes or water intake periods ?. It creates confusion because was not defined in M&M.

DISCUSSION

Why not presenting the discussion by item described in results ?. Readability could be improved by doing so.

CONCLUSIONS

Results should be separately stated followed by likely implications.

Comments on the Quality of English Language

Moderate editing of English language required

Round 2

Reviewer 1 Report

Comments and Suggestions for Authors

The paper is improved.

I am not sure why, however, the authors report that "a model with three dimensions only allow to visualize the overall P values and not the detailed comparisons". This can be done via emmeans package in R, similar packages in R, or similar ways in other software. Just need estimated marginal means and a post-hoc test to correct p-values for multiple tests. Usually a Bonferroni-Holm correction or similar. I can understand this was done before but the reasoning is not that it is not possible, as it is possible. Even with the current models, when you show the interaction effects you can actually plot the significant results as estimated marginal means. 

I would also remove part two from the title, not needed.

you were probably asked to add p-values in the discussion but it does not make sense, so please remove them. 

Comments on the Quality of English Language

na

Author Response

Reviewer 1:

I am not sure why, however, the authors report that "a model with three dimensions only allow to visualize the overall P values and not the detailed comparisons". This can be done via emmeans package in R, similar packages in R, or similar ways in other software. Just need estimated marginal means and a post-hoc test to correct p-values for multiple tests. Usually a Bonferroni-Holm correction or similar. I can understand this was done before but the reasoning is not that it is not possible, as it is possible. Even with the current models, when you show the interaction effects you can actually plot the significant results as estimated marginal means. 

Authors: Thank you for the comment. As also suggested by another reviewer, we rephrased new L 146 inn “As an ad-hoc decision, we used two models with two fixed factors each; climatic condition and trough type (Table 3) and climatic condition and cleaning status (Table 4) instead of a combined model with three possible fixed factors (climatic condition, trough type and cleaning status) because we had access to climatic data long time later to the first statistical analysis for the first published work [10].”

I would also remove part two from the title, not needed.

Author: To indicate the connection to the previous study, we were asked by the editor to include "part two" in the title. Therefore, we have kept "part two" in the title.

you were probably asked to add p-values in the discussion but it does not make sense, so please remove them. 

Author: As suggested, we removed the p-values in L 243, 262, 264, 266, 275, 289, 290, 292, 294, 302, 303, 312, 317, 318, 320

Reviewer 3 Report

Comments and Suggestions for Authors

GENERAL APPRAISAL

This manuscript is now in better shape. In my opinion, only minor details require attention.

TITLE, OBJECTIVES, RESULTS AND CONCLUSIONS

Much better matched

Format mistakes were addressed.

SIMPLE SUMMARY

L12-14: Livestock water requirements are expected to triple. Simultaneously, the fresh water availability declines while the risk of water borne diseases rises with increasing temperatures.

L16: Split “behavior in”.

ABSTRACT

OK

INTRODUCTION

OK

L50: Estrus status ?. What it means ?

L51: Space after period [8].Characterizing. Check all the away. There are several.

MATERIALS & METHODS

Much better.

L114 and 129: Tables 1 and 2 have titles above and below. Please adjust title of tables to format.

P4/L133-134: Compared or similar ?. In such a case, similar results ?.

L-141: the word “duration” is lost

L143-147: Here is a fundamental issue to solve: “These two models were used for clarity purposes since models with three dimensions only allow to visualize the overall P-values and not the detailed comparisons”. From my ignorant stand-point, I don’t think it is a valid argument because whenever you use the ”PDIFF statement in SAS”, it provides a complete comparison among each of the independent variables. However, I definitely think that this manuscript contains valuable information. Hence, perhaps more elegant ways to present it could be:

a) As an ad-hoc decision, we used two models with two fixed factors each; climatic condition and trough type (Table 3) and climatic condition and cleaning status (Table 4) instead of a combined model with three possible fixed factors (climatic condition, trough type and cleaning status) because we had access to climatic data long time later to the first statistical analysis for the first published work (Ref 10).

b) “As an ad-hoc decision, we used two models with two fixed factors each; climatic condition and trough type (Table 3) and climatic condition and cleaning status (Table 4) instead of a combined model with three possible fixed factors (climatic condition, trough type and cleaning status) because after including the climatic data we realized that degree of freedom and sample size were insufficient to produce a statistical comparison. And of course, run the statistical analysis with the 3 variable and verify this situation.  

RESULTS

OK

DISCUSSION

OK

CONCLUSIONS

OK.

Comments on the Quality of English Language

Minor editing of English language required

Author Response

GENERAL APPRAISAL

This manuscript is now in better shape. In my opinion, only minor details require attention.

Author: Thank you for reading the manuscript again and for your detailed comments. We tried our best to integrate them as completely as possible.

TITLE, OBJECTIVES, RESULTS AND CONCLUSIONS

Much better matched

Format mistakes were addressed.

SIMPLE SUMMARY

L12-14: Livestock water requirements are expected to triple. Simultaneously, the fresh water availability declines while the risk of water borne diseases rises with increasing temperatures.

Authors: As suggested, we rephrased L 12-13 in “Livestock water requirements are expected to triple. Simultaneously, the fresh water availability declines while the risk of water borne diseases rises with increasing temperatures”.

L16: Split “behavior in”.

Authors: As suggested, we split in new L 18 “behavior” and “in”.

ABSTRACT

OK

INTRODUCTION

OK

L50: Estrus status ?. What it means ?

Authors: We rephrased in new L52 “Water intake is a potential indicator detecting dairy cow’s health and heat“.

L51: Space after period [8]. Characterizing. Check all the away. There are several.

Authors: as suggested, we checked the entire manuscript for spaces. These changes were made in the clean version ("Manuscript R2 clean" document) to ensure that missing spaces were not overlooked by the backtracking mode.

MATERIALS & METHODS

Much better.

L114 and 129: Tables 1 and 2 have titles above and below. Please adjust title of tables to format.

Authors: Thank you for the hint, we adjusted it.

P4/L133-134: Compared or similar?. In such a case, similar results ?.

Authors: We deleted the repetition of the sentence in L178 “This type of evaluation can also be compared with that of a previously published paper [10]”

L-141: the word “duration” is lost

Authors: we rephrased new L 143-146 in “(i.e., total duration of drinking and tasting period, duration; number and duration of water intake periods and drinking breaks, sips per drinking episode)”

L143-147: Here is a fundamental issue to solve: “These two models were used for clarity purposes since models with three dimensions only allow to visualize the overall P-values and not the detailed comparisons”. From my ignorant stand-point, I don’t think it is a valid argument because whenever you use the ”PDIFF statement in SAS”, it provides a complete comparison among each of the independent variables. However, I definitely think that this manuscript contains valuable information. Hence, perhaps more elegant ways to present it could be:

  1. a) As an ad-hoc decision, we used two models with two fixed factors each; climatic condition and trough type (Table 3) and climatic condition and cleaning status (Table 4) instead of a combined model with three possible fixed factors (climatic condition, trough type and cleaning status) because we had access to climatic data long time later to the first statistical analysis for the first published work (Ref 10).
  2. b) “As an ad-hoc decision, we used two models with two fixed factors each; climatic condition and trough type (Table 3) and climatic condition and cleaning status (Table 4) instead of a combined model with three possible fixed factors (climatic condition, trough type and cleaning status) because after including the climatic data we realized that degree of freedom and sample size were insufficient to produce a statistical comparison. And of course, run the statistical analysis with the 3 variable and verify this situation.  

Author: Thank you for the comment. As suggested we rephrased new L 150 in “As an ad-hoc decision, we used two models with two fixed factors each; climatic condition and trough type (Table 3) and climatic condition and cleaning status (Table 4) instead of a combined model with three possible fixed factors (climatic condition, trough type and cleaning status) because we had access to climatic data long time later to the first statistical analysis for the first published work [10].”
